# Prospective Use of Probiotics to Maintain Astronaut Health during Spaceflight

**DOI:** 10.3390/life13030727

**Published:** 2023-03-08

**Authors:** Sahaj Bharindwal, Nidhi Goswami, Pamela Jha, Siddharth Pandey, Renitta Jobby

**Affiliations:** 1Amity Centre of Excellence in Astrobiology, Amity University Mumbai, Mumbai 410206, Maharashtra, India; 2Department of Biology, University of Naples Federico II, 80131 Naples, Italy; 3Amity Institute of Biotechnology, Amity University Maharashtra, Mumbai 410206, Maharashtra, India; 4Sunandan Divatia School of Science, NMIMS University Mumbai, Mumbai 400056, Maharashtra, India

**Keywords:** astronaut, spaceflight, probiotics, microbiome, simulated microgravity (SMG)

## Abstract

Maintaining an astronaut’s health during space travel is crucial. Multiple studies have observed various changes in the gut microbiome and physiological health. Astronauts on board the International Space Station (ISS) had changes in the microbial communities in their gut, nose, and skin. Additionally, immune system cell alterations have been observed in astronauts with changes in neutrophils, monocytes, and T-cells. Probiotics help tackle these health issues caused during spaceflight by inhibiting pathogen adherence, enhancing epithelial barrier function by reducing permeability, and producing an anti-inflammatory effect. When exposed to microgravity, probiotics demonstrated a shorter lag phase, faster growth, improved acid tolerance, and bile resistance. A freeze-dried *Lactobacillus casei* strain Shirota capsule was tested for its stability on ISS for a month and has been shown to enhance innate immunity and balance intestinal microbiota. The usage of freeze-dried spores of *B. subtilis* proves to be advantageous to long-term spaceflight because it qualifies for all the aspects tested for commercial probiotics under simulated conditions. These results demonstrate a need to further study the effect of probiotics in simulated microgravity and spaceflight conditions and to apply them to overcome the effects caused by gut microbiome dysbiosis and issues that might occur during spaceflight.

## 1. Introduction

Human space exploration has increased recently as more missions are planned by various international space agencies. The National Aeronautics and Space Administration’s “Human Research Program” is currently planning long-term human spaceflight missions to Mars and the Moon. Various studies indicate that astronauts find it difficult to maintain their health and face many health issues during short and long space flights due to exposure to multiple stressors such as microgravity and radiation. Thus, it is necessary to understand human health risks related to space travel. Astronauts who spend 6–12 months aboard the International Space Station (ISS) have experienced changes in the gut microbiota and different physiological changes. These changes include genitourinary tract infections, cardiovascular issues, changes in resistance and virulence of bacteria, changes in immune response, and the development of cancers due to exposure to radiation [1,2]. It is crucial to take the necessary precautions to preserve the astronauts’ health as space missions last very long periods of time [3].

The World Health Organization has characterized probiotics as “Live microorganisms that, when administered in adequate amounts, confer a health benefit on the host” [4]. Consuming a few probiotic strains has been shown to regulate the immune system and intestinal flora, leading to an increase in good bacteria such as *Lactobacilli* and *Bifidobacteria* and a decrease in harmful microbes. Probiotics such as *Lactobacillus casei* strain Shirota (LcS) can improve innate immunity and increase natural killer cell activity by primarily enhancing the production of interleukin-12 by monocytes and macrophages. LcS, upon ingestion, reaches the intestinal microbiome in live form and maintains the intestinal microbiome [5,6]. Probiotics have been shown to influence neuroactive substance synthesis and release. *Lactobacillus acidophilus* has been shown to modulate the expression of cannabinoid receptors [7]. As a potential probiotic that can make good use of gastrointestinal mucin, *Akkermansia muciniphila* is inextricably linked to host metabolism and immune response. It has the potential to be a therapeutic target in microbiota-related diseases such as colitis, metabolic syndrome, immune diseases, and cancer [8]. As a result, a study suggests that next-generation probiotics derived from *Akkermansia* may reduce the risk of diseases associated with chronic inflammatory [7]. Oral administration of the prominent gut microbe *Faecalibacterium prausnitzii* has recently been discovered to show anti-inflammatory properties by increasing the production of IL-10 (a cytokine) and tumor necrosis factor (TNF) in the colon to improve intestinal disease [9]. Another study also shows the anti-inflammatory potential of *Lactobacillus bulgaricus* and *Streptococcus thermophilus* strains isolated from Bulgarian homemade yoghurt. *Lactobacillus*, *Bifidobacterium*, and *Streptococcus* probiotic strains are mainly used to prevent or treat oral infections [10].

Certain gut microbiota plays an important role in nutritional functionality and contributes to vitamin availability and short-chain fatty acid production. Gut microorganisms can produce vitamin B12, vitamin K, pyridoxine, folate, biotin, nicotinic acid, and thiamine [11]. Plaque or dental biofilms in the buccal cavity cause poor oral health; however, lactic acid bacteria (LAB) interact with that biofilm/plaque and through antimicrobial activity, destroy the causative agents [12].

During a long-term spaceflight, the reliability of efficient health management is essential. According to research, spaceflight causes changes in human physiology [13]. These changes can be of various natures: physiological including gastrointestinal distress, dermatitis, and respiratory infections; immunological [14] and microbiome [15]. Studies on Earth have shown probiotics to be beneficial in the improvement of health issues that are faced during spaceflight. They aid by competing with pathogens, reducing gastrointestinal issues, strengthening tight junctions in intestinal epithelial cells, producing essential metabolites, and interacting with host cells to promote physiological and immune changes [16,17,18]. This review focuses on various spaceflight problems that astronauts encounter and how consumption of probiotics can help to alleviate these problems, which could aid astronauts in overcoming spaceflight difficulties.

## 2. Health Issues during Spaceflight

Space is a harsh environment, and advances in material science, power generation, robotics, and medical requirements are essential to ensure astronauts’ survival during settlements and interplanetary journeys. The emerging field of bioastronautics aims to address some of the medical issues that astronauts face while in space. Because of the hostile environment in space, astronauts face several health risks during both long- and short-duration spaceflight [19,20]. A diagrammatic representation of health issues faced by astronauts during spaceflight is shown in (Figure 1).

### 2.1. Changes in the Microbiome

Joshua Lederberg instituted the term “human microbiome” in 2001. He characterized it as the “Natural network of commensal, symbiotic and pathogenic microorganisms that genuinely share our body space.” The human microbiome consists of various advantageous symbionts, primarily bacteria, that actively boost health. With a change in the microbiota, the increase in pathogens can affect homeostasis and cause different diseases. In both long and short-term space missions, changes in the gut, nasal, and oral bacterial profiles of astronauts have been observed. These progressions are related to a diminishing in the overall wealth of advantageous microbes from the genera *Lactobacillus* and *Bifidobacterium* and an expansion in opportunistic microorganisms, for example, *Escherichia coli*, *Clostridium* sp., *Staphylococcus aureus*, *Fusobacterium nucleatum*, and *Pseudomonas aeruginosa* [3].

An investigation of the microbiota from nine astronauts who spent a year aboard the ISS presents proof demonstrating a change in the microbial population of the gastrointestinal (GI) tract, nose, tongue, and skin during space missions. The DNA collected from the microbial samples in the study was subjected to 16S rRNA gene sequencing to determine the microbial makeup. This study distinguished the space-related increase in *Parasutterella* sp. number. It is categorically linked to chronic intestinal aggravation in people with inflammatory bowel disease. The study also found a space-related decrease in the population of three bacterial genera with anti-inflammatory properties: intestinal *Fusicatenibacter*, *Pseudobutyvibrio*, and *Akkermansia*. Fewer inflight changes were encountered in the nasal microbiota [1,6].

The study by Liu et al. [15] was the first to show the effect of short-term spaceflight missions on the human gut microbiome. The study showed that *Bacteroides* abundance increased after spaceflight, with a decrease in *Lactobacillus* and *Bifidobacterium*. *Bacteroides* efficiently degrade dietary fibres in the human gut and are essential phenolic acid and propionate producers. The *Bacteroides* genus has several pathogenic bacterial species that reproduce rapidly under stressful conditions. The space environment weakens the human immune system, and the number of *Bacteroides* can increase. In the human gut, *Lactobacilli* are responsible for a large amount of lactic acid production. Additionally, studies demonstrate that *Bifidobacterium* generates lactic and acetic acids from sugars. A decline in the population of *Lactobacillus* and *Bifidobacterium* species can interfere with the host immune system’s functioning in the gut and the working of the gut microbiota, and spaceflight-affected immune systems can cause latent viruses’ reactivation and an increase in opportunistic pathogens’ number in the gut.

The twin study also showed that metabolites, such as 3-indole propionic acid, which has anti-inflammatory effects, were noted at low levels in flight throughout the study. The following study also stated a change in the microbiome functioning in the flight subject compared to the ground subject concerning microbial communities [21].

A study by Siddiqui et al. [22] used a hindlimb unloading (HU) mouse model on the ground to simulate microgravity conditions to investigate changes in the gut microbiota bacterial composition. They found that hindlimb unloading causes changes in the gut microbiota, including a decrease in the diversity of useful gut microbiota which can lead to increased permeability and gut inflammation. The study showed a decrease in *Akkermansia muciniphila*, *Eubacterium coprostanoligenes*, and *Burkholderiales* in mice exposed to simulated microgravity compared to normal mice. These bacterial genera are associated with anti-inflammatory properties, gut homeostasis, and health benefits such as kidney stone prevention. The study also highlighted the importance of a balanced proportion of *Firmicutes* and *Bacteroidetes* in maintaining overall health, with modifications in their proportions leading to dysbiosis and associated health issues. *Firmicutes* play a role in host metabolism and nutrition, while *Bacteroidetes* are associated with immunomodulation. These results suggest that changes in the gut microbiota may contribute to negative health effects experienced in spaceflight [22].

The analytical tool “Similarity Test for Accordant and Reproducible Microbiome Abundance Patterns” or STARMAPs tests the similarity in two-space research datasets for finding microbiota variations. The study also discovered that spaceflight-related microbiota changes during the RR-1 (Rodent Research 1) and STS-135 (Human Mission) missions were similar, implying a vigorous change in mammalian gut microbiota due to spaceflight [23]. The effect of microgravity on the gut microbiota of the astronauts during spaceflight is also summarized in Table 1.

#### 2.1.1. The Microbiota–Gut–Brain Axis and Its Relation to the Mental Health of Astronauts

The gut microbiota is comprised of 10⁶ viral, bacterial, and protozoa cells, making it the most numerous community of human microbiota. The brain influences microbiota function and composition by altering intestinal permeability. The brain, via the autonomic nervous system (ANS), may also influence immune function. The gut–brain axis (GBA) is a bidirectional communication pathway between the central and enteric nervous systems. It links the emotional and cognitive centers of the brain with peripheral intestinal functions. Recent research advances have highlighted the importance of the gut microbiota in influencing these interactions [24].

The composition of the gut microbiota undergoes many transitions during the life span that parallel dynamic periods of brain development, aging, and maturation. Dysbiosis can result from environmental factors such as diet, gravity, stress, and radiation. Additionally, the gut microbiota plays an important role in the development of both adaptive and innate immune responses. The gastrointestinal microorganisms act as the relay stations of information aggregation and transmission to specific areas of the brain, especially the cerebellum, affecting mental health. The influence of environmental factors related to space travel converge on these microorganisms [25].

#### 2.1.2. Mental Health Challenges Faced by Astronauts during Spaceflight

Spaceflight is a unique and difficult environment that can have a substantial influence on astronauts’ mental health [26]. Isolation, confinement, microgravity, disruption of circadian rhythms, and communication delays can all contribute to psychological stress, anxiety, depression, and other mental health issues [27,28]. Research shows that astronauts are more likely to be emotional and have mental disorders when they are in space [29]. The lack of social support and isolation that happens during spaceflight is one of the leading reasons for mental health difficulties for astronauts. Astronauts are isolated from their families and friends for lengthy periods, making it difficult for them to engage in regular social engagements. This might result in feelings of isolation, boredom, and psychological anguish [28]. Another major challenge is the disturbance of the sleep–wake cycle caused by persistent artificial light exposure and the absence of a natural day–night cycle in space [30]. A recent study by Ma et al. [31] also showed the connection between the probiotic-induced gut microbiome and alleviated stress levels in adults, thus proving the role of the gut–brain axis in reducing the effects of stress.

Mental health is an important part of spaceflight, and astronauts are more likely to experience psychological stress, anxiety, and other mental health issues. It is critical to continue exploring and adopting techniques to improve astronauts’ mental health while in space.

### 2.2. Genitourinary Tract Infection

On long-term space missions, astronauts face many physical challenges that could affect their genitourinary health. During spaceflight, the microgravity environment causes a decrease in plasma volume and an increase in urine output due to fluid shifts from the lower extremities towards the upper body, leading to the risk of urinary tract infections (UTIs) due to decreased urine flow and urinary stasis. Additionally, stress and altered sleep–wake cycles during spaceflight may contribute to an increased incidence of nocturia [33].

Male astronauts may face decreased testosterone levels due to radiation exposure during spaceflight, leading to reduced libido and erectile dysfunction. On the other hand, female astronauts may experience menstrual irregularities and pelvic congestion syndrome, causing pain, pressure, and discomfort in the pelvic region [33,34]. UTIs are common issues in space, with female astronauts experiencing a greater frequency than male astronauts. Uropathogens such as *Staphylococcus saprophyticus* and *Escherichia coli* are significant causes of all urinary tract infections because of their capacity to attach to uroepithelial cells through adhesion molecules. In vitro investigations have shown that both pathogenic and nonpathogenic *E. coli* strains exhibit better adhesion and invasion under microgravity. This higher adhesion, together with *E. coli*’*s* accelerated growth kinetics in space, may be responsible for disease progression [3].

### 2.3. Virus Reactivation in Spaceflight 

A study by Sonnenfeld & Shearer [35] elucidated immune system compromise, a possible development of the malignant condition, and latent virus reactivation infection in humans during spaceflight. During spaceflight, a significant number of these symptoms are related to a weakening of the immune system due to the reactivation of two viruses: the Epstein–Barr virus and the Varicella–Zoster virus [1]. Latent virus reactivation is a biomarker for the status of the immune system of astronauts, and factors contributing to it are an increase in glucocorticoid secretion, change in cytokine production, and decreased function of immune cells targeted at eliminating the viruses. The presence of viral DNA in bodily fluids indicates viral reactivation [36].

### 2.4. Resistance of Bacteria and Changes in Bacterial Virulence

Zhang et al. [37] studied changes in the antimicrobial resistance of the *S. enteritidis* strain, which was carried to space by the Shenzhou-11 spacecraft. Compared with the ground strain, the flight strain showed enhanced amikacin resistance, an increased growth rate, and some metabolism changes. *Escherichia coli* MG1655 revealed antibiotic resistance when exposed to long-term low-shear modelled microgravity (LSMMG) and background antibiotics such as chloramphenicol, cefalotin, tetracycline, cefoxitin, cefuroxime, and cefoxitin. The strain showed resistance to chloramphenicol and cefalotin for more than 110 generations, even after the elimination of the LSMMG environment and trace antibiotic exposure. The adapted strain of *Escherichia coli’s* genome sequence showed about 25 changes. These genomic changes were associated with antibiotic resistance, with a change in four antibiotic resistance genes: *ompF*, *acrB*, *mdfA*, and *marR* [38]. According to a study by Liu et al. [15] spaceflight changes the virulence of bacteria. When examining virulence genes, they discovered that the gut microbiome had an impact on some virulence factors (VF). An example of such a change was an increase in factor VF0367, linked with lipopolysaccharide development, which forms a protective layer in *Brucella* [15]. A recent study showed that the number of readings for the Streptomyces EF-Tu mutation marker increased significantly after the astronaut’s journey. This marker identifies elfamycin resistance-inducing sequence variations of the *Streptomyces cinnamoneus* elongation factor Tu. The increase in EF-Tu mutations after spaceflight shows that elfamycin resistance may have increased due to spaceflight circumstances [39]. These studies suggest increased pathogenicity of certain microbes after exposure to spaceflight.

### 2.5. Epithelial Barrier Disruption and Inflammatory Bowel Disease (IBD)

Inflammatory bowel disease (IBD) is a chronic, recurrent inflammatory condition of the gastrointestinal tract marked by epithelial barrier disruption and immunological dysregulation. Recent research has found that astronauts endure gastrointestinal discomfort, including IBD-like symptoms, when in space, most likely as a result of the impact of microgravity on the intestinal epithelium. Changes in tight junction (TJ) proteins produce epithelial barrier disruption, which leads to increased intestinal permeability and the subsequent translocation of luminal antigens across the epithelium [40,41]. An alteration in the expression or localization of TJs may lead to a leaky gut condition due to increased permeability to molecules diffusing from the lumen to the lamina propria [42]. In their study, Alvarez et al. [43] found a delay in the localization of TJ proteins—occludin and ZO-1 under simulated microgravity conditions. The findings indicate that simulated microgravity damaged the epithelial barrier and an underlying susceptibility to the barrier persisted even after the microgravity condition was removed. This underlying barrier disruption makes astronauts prone to various intestinal epithelial cell barrier defect diseases such as Crohn’s disease, ulcerative colitis, celiac disease, and type I diabetes [44].

IBD is reported in astronauts during spaceflight with an increased intestinal paracellular permeability as a result of TJ protein disruption [1,45]. A study reported a reduced expression and distribution of TJ proteins such as occludin, claudin-1, claudine 04, and JAM-A, and an increase in the expression of claudin-2 [46]. Another study by Yi et al. [47] suggested that *Lactobacillus reuteri* LR1 can treat intestinal disorders associated with impaired function of the epithelial barrier. The infection with enterotoxigenic *E. coli* K88-induced an increase in permeability of IPEC-1 cell monolayers. The probiotic LR1 significantly improved the epithelial barrier function, and reduced adhesion and colonization by coliforms.

### 2.6. Immunological Alteration during Spaceflight

Astronauts face the issue of immune cell alteration during spaceflight. Innate immunity, or the first line of defence, plays a vital role in prolonging healthcare among astronauts. Immunological changes observed in astronauts during space flight have been shown in (Table 2). A study conducted at Johnson Space Center, Houston, showed an 85% increase in neutrophils during a 5–11-day spaceflight mission as compared to pre-flight levels along with remarkably lower values in phagocytosis [48]. An increase in the number of white blood cells, polymorphonuclear leukocytes, was also observed in short-duration spaceflight missions to the ISS [49]. Similar effects have been observed in astronaut long duration spaceflight missions. An increase in the level of white blood cells [14]. Another study by Makedonas et al. [50] reported an increased inflammation in the astronaut during 1-year NASA “twins” study aboard the International Space Station. Cosmonauts on a long duration (>140 days) spaceflight have shown an increased release of endocannabinoids combined with immune activation, which mimics the risk of inflammation-related disorders in humans. The increased inflammation persisted for 30 days post-flight [51]. The alterations in gravity experienced by astronauts can also impact the microenvironments of two critical primary lymphoid organs, the thymus and bone marrow. These organs are responsible for generating lymphocytes or white blood cells. Changes in lymphocyte production can have an indirect influence on acquired immune responses, altering how the immune system typically reacts to inflammation, infections, and tumors [52]. 

### 2.7. Changes in Cardiovascular Functions

Astronauts face weightlessness in space, which results in a redistribution of bodily fluids to the thoracic–cephalic areas from the lower half of the body. This fluid transfer is responsible for a cardiovascular deconditioning syndrome characterized by hypotension, the possibility of presyncope or syncope, and a reduced stress capability [57].

Astronauts experience metabolic stress while in space. Metabolic stress is a strong predictor of both heart disease and Type 2 diabetes [58]. Spaceflight also poses the risk of malignant arrhythmias development, as changes caused during spaceflight underline an augmented repolarization heterogeneity. Further studies are needed to understand the physiological changes occurring in the body, which will also help provide deeper insights into changes in human health upon commercialization of spaceflight [59].

### 2.8. Effect of Cosmic Radiation on Astronauts

Humans are exposed to space radiation while in space. These are the galactic cosmic rays generated outside our solar system, solar particles released from the sun, and the radiation confined due to the Earth’s magnetic field. These space radiations endanger astronauts because they cause several types of cancer. Female astronauts have a 20% higher chance of getting cancer than male astronauts. This is mostly because breast and ovarian cancers are more common in women. During travel, people can experience short-term effects such as changes in their blood, diarrhoea, nausea, and vomiting [3]. Radiation reduces the diversity of intestinal flora and alters the composition of the gut microbiota [60]. Reports from previous Apollo, Skylab, and Russian modular space station (MIR) flights suggest that astronauts saw flashes of light moving across their visual field, presumably due to a change in perception produced by ionizing radiation, showing that visual disturbances also seem to be associated with radiation exposure [61]. During a 6-month mission to the ISS, Moon, and beyond, an astronaut is exposed to radiation of around 50–2000 millisieverts (mSv). A radiation dose above 100 mSv is documented to cause cancer [62]. The STARMAPs statistical analysis study showed that the spaceflight-related microbiota changes compared with the space-like radiation-induced changes on the ground were different. They suggested the difference could be because the ISS is in lower orbit inside the Van Allen belt. Hence, the research subjects in the study were not exposed to cosmic radiation. The study proves that understanding space radiation far from Van Allen Belts is vital in the near future [23].

Pro-inflammatory reactions to weightlessness, radiation, stress-induced hyperthermia, or a combination of these factors during spaceflight can cause “space fever,” which can affect astronauts’ health and energy, nutrient and fluid requirements, and physical and cognitive performance during long-duration spaceflight [63].

## 3. Probiotics and Their Role in Space Biology

Space exploration has urged scientists to develop and plan human-crewed missions to the Moon and Mars. Such long-duration missions require extensive knowledge of how space travel affects the astronauts’ health. The advent of Apollo 11 and various simulation experiments on Earth and on the ISS have allowed us to understand how space affects microbes and humans. As mentioned in Section 2.1, the maintenance of the human gut microbiome is an essential aspect of long-duration space travel. Imbalances in the gut microbiome have caused many diseases, and space travel has been shown to cause changes in the gut microbiome. Probiotics can help with GI issues such as acute infectious diarrhoea, *Helicobacter pylori* infection, antibiotic-associated diarrhoea, irritable bowel syndrome, ulcerative colitis, and constipation, as well as improve gut barrier functions [64,65]. Probiotics also help maintain the immune system, prevent cancer, and help with psychological issues [66]. The most widely used probiotics include members from the *Lactobacillus, Bifidobacterium,* or *Saccharomyces* species [67]. We see further how probiotics can be a beneficial supplement (Table 3).

### 3.1. General Mechanism of Action of Probiotics

#### 3.1.1. Inhibition of Pathogen Binding

Probiotic strains inhibit pathogen binding to the epithelial layer by altering mucus secretion levels. Probiotics can improve the strength of the intestinal barrier by increasing the number of goblet cells (secrete mucin) that support the mucus layer. The mucus layer has a role in diminishing the binding of pathogenic bacteria to mucosal epithelial cells, and probiotics work by inducing mucus secretion [68,69]. Otte and Podolsky [70] found that *Lactobacillus* strains changed the way MUC2, MUC3, and MUC5AC were expressed in HT29 cells. Probiotic strains can also inhibit pathogen binding to the epithelial layer by competing for the adhesion site. Human mucus-binding pili make it possible for some probiotics to colonize the body better [71]. Probiotics compete for lectin-binding sites on glycoconjugate receptors present on the microvilli surfaces of epithelial cells [72,73]. *L. plantarum* and *Lactobacillus rhamnosus* strain GG have been shown to inhibit the attachment of pathogenic *E. coli* to the epithelium [74].

#### 3.1.2. Use of Probiotics for Intestinal Disorders

The pathogenesis of irritable bowel syndrome (IBS) may involve altered gut immune activation, gut microbiome dysbiosis, altered brain–gut axis, and increased intestinal epithelial cell permeability [75]. Probiotics influence symptoms involved in IBS, such as bloating, flatulence, altered bowel movements, gut microbiota dysbiosis, and abdominal pain [76]. Probiotics act by inhibiting pathogen adherence, enhancing epithelial barrier function by reducing its permeability, and producing an anti-inflammatory effect [77]. The integrity of GIT is maintained by epithelial cells, which serve as a barrier between the host immune system and the external environment. In the probiotic *Escherichia coli* strain Nissle 1917 (EcN), an overriding signalling effect leads to the restoration of disrupted epithelial cells. This makes probiotic EcN more effective at treating inflammatory bowel disease [41]. The probiotic Lactobacillus plantarum MB452 also improves the integrity of the intestinal barrier by increasing the expression of tight junction proteins—cingulin and occludin. These proteins help maintain the repair of epithelial cells [67]. *Bifidobacterium* sp. is another group of probiotics that help preserve tight junctions’ integrity in the GI mucosa. They protect the epithelial barrier from acute colitis by preventing the redistribution of occludin and TJ proteins [78].

#### 3.1.3. Immune System Maintenance

Probiotics can modulate the immune system mainly by (1) altering immunoglobulin/ cytokine secretion, (2) strengthening the epithelial gut barrier, (3) increasing macrophages or natural killer cells activity, (4) competitively binding to the epithelial layer preventing pathogenic microbes from binding, and (5) modulating the secretion of mucus. Antigenic particles produced by probiotics, not whole bacteria, can enter epithelial cells and contact immune cells [79].

Few probiotic strains, such as *Lactobacillus rhamnosus* GG and *Bifidobacteria*, modulate cytokine production from various cell types, altering mucosal and systemic innate and adaptive immune responses [80]. Probiotics interact with epithelial cells and modulate cytokine release by altering cellular signal transduction pathways [81].

Different probiotic strains act by stimulating the production of different immune system components. These include the stimulation of IL-10 and IL-20 production by mononuclear cells in lactic acid bacteria [82], induction of IL-6 production in *Lactobacillus rhamnosus* GG [80], and prevention of cytokine-induced apoptosis and inactivating activation of pro-apoptotic p38 mitogen-activated protein kinase by TNF, IL-1a, or gamma-interferon in *Lactobacillus rhamnosus* GG [83], suggesting the increased survival of intestinal cells [79].

Given the impacts on immunity, using probiotics to promote SCFA formation would therefore boost nutritional and metabolic resources as well as lymphocytes’ capacity for virus elimination, potentially reducing the re-emission of latent viruses [84].

#### 3.1.4. Antimicrobial Activity of Probiotics

Other mechanisms by which probiotics inhibit microbial growth include the synthesis of organic acids, toxic substances, and bacteriocins [85]. Lactic acid bacteria (LAB), propionic acid bacteria, and *Bifidobacteria* have been used in the preservation and fermentation industries for centuries. The factors making them efficient for use in preservation can be attributed to low pH, reduced amounts of carbohydrates, and the production of antimicrobial compounds. These bacteria can produce antimicrobial substances, making them the right candidate for selection as a probiotic [86].

LAB produces organic acids such as acetic acid, lactic acid, and propionic acid through the fermentation of glucose. Lactic acid and acetic acid have an inhibitory effect on yeast, moulds, and bacteria [87]. In addition to increased pH, the undissociated acid diffuses over the cell membrane. It dissociates, releasing H+ ions in the cytoplasm, causing a collapse in the electrochemical gradient and the subsequent bacteriostasis or death of bacteria [88].

Bacteriocins produced by LAB are antimicrobial peptides synthesized by ribosomes [89]. Bacteriocins primarily target the cell membrane, inhibit spore germination, cause the inactivation of anionic carriers, and alter enzymatic activity with a bacteriostatic or bactericidal effect depending on the sensitivity of the cell. These peptides are usually effective on closely related bacterial species and Gram-positive bacteria [90].

#### 3.1.5. Probiotics Used for Antibiotic-Associated Diarrhoea

The gut microbiota is subjected to change during spaceflight, and antibiotics are used for treatment [1]. Although antibiotics are crucial to eradicating bacterial infections, they cause substantial damage to the microorganisms in the gut microbiota [91]. Antibiotic usage can cause various problems, such as colonization by pathogenic *Clostridium difficile*, which causes chronic gastrointestinal tract issues and extreme diarrhoea. In normal conditions, *C. difficile* faces competition by commensal bacteria in the GI tract, but when the gut microbiota is compromised (as observed during space travel), *C. difficile* can colonize the tract [92]. Probiotics can be used to replenish the GI microbiota, and they can also be used to treat *C. difficile* infections [93]. Probiotics can be used to treat antibiotic-related diarrhoea.

#### 3.1.6. Probiotics as Prophylaxis for Cancer

Due to radiation exposure, astronauts have a higher likelihood of cancer. Consumption of soy milk fermented with probiotics acts as a prophylactic measure against breast cancer through the anti-estrogenic effect of isoflavones [94]. The development of colon cancer depends on several factors. Evidence has shown a correlation between alterations in the makeup of the gut microbiome and the development of colorectal cancer. Probiotics may affect how the immune system and the gut microbiota communicate and may help prevent colorectal cancer [95]. Kefir (fermented milk with probiotics) contains bioactive compounds, such as polysaccharides and peptides, which can inhibit proliferation and apoptosis induction in tumour cells. Studies have revealed that kefir can act on colorectal and breast cancer [96].

#### 3.1.7. Probiotics for Stress/Anxiety

Maintaining astronauts’ emotional and physical condition is a crucial factor for future long-duration space missions. Stress is undoubtedly one of the most worrisome agents that can affect the crew’s overall well-being due to its effects on human health and performance [97]. Anxiety and stress have been linked to gut dysbiosis. The study by Ma et al. [31] reported that the intake of *Lactobacillus plantarum* P-8 ameliorated anxiety/stress symptoms in humans. It was also discovered that probiotic consumption enriched the gut-derived metabolite gamma-aminobutyric acid (GABA) synthesis pathway by *Bifidobacterium adolescentis*, GABA and histamine are important neurotransmitters that travel through the vagus nerve to the gut-brain axis.

#### 3.1.8. Probiotics for Urinary Tract Infection

Probiotics such as *Lactobacillus rhamnosus* GR-1 and *Lactobacillus reuteri* RC-14 have anti-infective properties, which have been tried in females and appeared to forestall UTIs to a comparable degree to the long-haul, low-portion antimicrobials without the reactions [3].

**Table 3 life-13-00727-t003:** Probiotics: mechanism of action and their health benefits.

Sr. no	Probiotics	Mechanism of Action	Primary Outcome of the Probiotic	Reference
1	*Lactobacillus rhamnosus* GG and *L. plantarum*	Inhibition of pathogen binding	Has been shown to inhibit the attachment of pathogenic *E. coli* to the epithelium	(Wilson and Perini et al., 1988), [74]
2	*Escherichia coli* strain Nissle 1917 (EcN) and *Lactobacillus plantarum* MB452	Intestinal disorders	*Escherichia coli* strain Nissle 1917 (EcN) restoration of disrupted epithelial cells *Lactobacillus plantarum* MB452 enhances intestinal barrier integrity	(Zyrek et al., 2007) and (Ulluwishewa et al., 2011), [41,67]
3	*Lactobacillus rhamnosus* GG	Immune system maintenance	Induction of IL-6 production	(Yan and Polk, 2011), [80]
4	Lactic acid bacteria and *Bifidobacteria*	Antimicrobial activity	By synthesis of organic acids, toxic substances, and bacteriocins	(Bermudez-Brito et al., 2012) and (Dunne at al., 2001), [85,86]
5	Kefir	Prevention of cancer	Production of bioactive compounds which can inhibit proliferation and apoptosis induction in tumour cells	(Sharifi et al., 2017), [96]
6	*Lactobacillus rhamnosus* GR-1 and *Lactobacillus reuteri* RC-14	Prevention of urinary tract infections	Anti-infective properties	(Urbaniak and Reid, 2016), [3]

#### 3.1.9. Short-Chain Fatty Acids and Their Role in Gut Microbiota Maintenance

Probiotics can produce short-chain fatty acids (SCFAs) [98]. SCFAs are organic by-products of fermentation. These are produced in the lumen of the intestine when non-digestible carbohydrates are incompletely broken down in the anaerobic environment by the gut microbiota. SCFAs are mainly composed of acetate, butyrate, and propionate [99,100]. SCFAs have a vital part in the regulation of the immune system. The maintenance, structure, and production of intestinal mucus are dependent on the gut microbiota and diet. A fibre-rich diet leads to the production of SCFAs by the gut microbiota, which improves mucus and antimicrobial peptide production and a higher expression of TJ proteins. A fibre-deficient diet results in altered gut microbiota, leading to a drop in the mucus layer and increasing susceptibility to infections and chronic inflammatory diseases [100]. SCFAs are also signaling molecules that regulate the formation of interleukin-18 by binding to the GPR41 and GPR43 receptors of gut epithelial cells and immune cells [101].

A study by Silva et al. [102] has reported SCFAs may have a direct effect on the brain by supporting blood-brain barrier (BBB) integrity, modulating neurotransmission, influencing neurotrophic factor levels, and promoting memory consolidation. A study reported SCFA butyrate enhances intestinal barrier function. Adenosine monophosphate-activated protein kinase (AMPK) upon activation facilitates the tight junction assembly and regulates the metabolic pathways in fatty acid metabolism and protein synthesis [103]. 

MARS 500 was a six-month ground-based experiment that included a faecal examination of six crew members. The results showed a continual variation in the relative abundance of butyrate-producing *Faecalibacterium prausnitzii* and *Roseburia faecis* in the gut microbiota of all crew members. This indicates a change in SCFA production and possible ramifications for supporting the microbiota–host mutualistic relationship [7].

Lunar Palace 1 is another experiment that was carried out on the ground. Three crew members were used to test how well the Bioregenerative Life Support System (BLSS) worked. They consumed a high-fibre diet and followed a fixed timetable that included substantial manual work in the plant cabin. The results showed similar changes in the gut microbiota composition in crew members with a high diversity and number of *Lachnospira, Faecalibacterium,* and *Blautia* microorganisms. This also stipulates that a high-fibre diet and lifestyle might be beneficial for the support of healthy gut microbiota [32].

### 3.2. Microgravity/Simulated Microgravity Studies on Probiotics

For a probiotic to be effective, it should have some specific characteristics. Some of these are stability against acid and bile, adherence to human intestinal cells, antagonism against enteric pathogens, and the production of antimicrobial substances. However, these characteristics can change based on environmental factors and microgravity. Several studies have been carried out to test probiotics in in vitro and in vivo conditions to understand their potential health benefits and safety for astronauts’ health during spaceflight. We summarize some of these studies in the following paragraphs.

A study conducted by Shao et al. [104] to investigate the impact of simulated microgravity conditions on *Lactobacillus acidophilus* revealed a considerable effect on some biological activities and features. The key findings were (1) no significant change in *L. acidophilus* morphology, (2) shortened lag phase, (3) increased growth rate, (4) enhanced tolerance to acids (pH 2.5) with resistance to bile, (5) decreased sensitivity to sodium penicillin, cephalexin, and sulphur gentamicin, (6) no significant change in *L. acidophilus* adhesion ability, and (7) increased antimicrobial activity against *S. aureus* and *S. typhimurium*. These alterations caused by simulated microgravity (SMG) on *L. acidophilus* probiotics can be advantageous to astronauts during spaceflight. These probiotics can tolerate stressful conditions and persist for a longer duration in the GI tract. Because there is no change in its adherence ability, it can help maintain intestinal epithelial barrier function and prevent pathogens from entering [103].

In another study, Senatore et al. [105] examined the *Lactobacillus reuteri* for its metabolism and gene expression in SMG conditions. They found no changes in bacterial growth, cell size, and shape with respect to the control. On the other hand, increased tolerance for the GI passage and enhanced production of the bioactive compound reuterin were seen [32].

The freeze-dried *Lactobacillus casei* strain Shirota capsule (LcS) was tested for its stability on the ISS for a month. LcS capsules from spaceflight did not differ in genetic profiles, growth pattern, carbohydrate fermentation, reactivity to LcS-specific antibody, and cytokine-inducing ability with respect to the control samples kept at the ground-based lab. LcS has been shown to enhance innate immunity and balance intestinal microbiota and can be used to combat the immune problems associated with spaceflight [6,44].

### 3.3. Commercial Probiotics’ Shelf Life and Survival in a Simulated Gastrointestinal Tract

Three commercial probiotics, namely *Lactobacillus acidophilus* strain DDS-1, *Bifidobacterium longum* strain BB536, and spores of *Bacillus subtilis* strain HU58 were tested for survival under conditions expected to be encountered during a 3-year round trip to Mars. The parameters tested were survival to:Long-term storage at ambient conditions;Simulated galactic cosmic radiation and solar particle event radiation;Exposure to simulated gastric fluid;Exposure to simulated intestinal fluid.

According to the study, radiation exposure had little impact on the probiotic strains examined. However, the shelf lives and survival rates of the three strains differed significantly during simulations of their passage through the upper gastrointestinal tract. According to the findings, only *Bacillus subtilis* spores could survive in all conditions. This suggests that probiotics made up of bacterial spores may be a viable choice for long-term human space travel [106].

## 4. Conclusions

For long-term space travel, a crucial factor is maintaining an astronaut’s health. Various physiological changes in the health of the flight crew have been observed, which include such things as a change in the gut microbiome leading to an altered MGB axis effect on mental health, genitourinary tract infections, reactivation of the virus, the resistance of bacteria and changes in virulence, the lowering of immunity and changes in immune response, cardiovascular issues, and the development of cancer due to radiation exposure. This review attempts to understand the possible use of probiotics, which can be used to tackle these health issues caused by spaceflight. Maintaining the gut microbiome is important for long-term space travel, and many diseases are caused by changes or imbalances in the gut microbiome. Due to the known benefits of probiotics for the gut microbiome and overall health, their use as a dietary supplement or as an addition to food during spaceflight could be a promising alternative to counteract the dysregulations and health consequences experienced by space travellers. However, experiments performed on probiotics in simulated microgravity conditions do not entirely mimic long-term space travel. More studies need to be conducted on probiotics to validate their use in space, check their efficacy as countermeasures for the health issues mentioned above, and change the properties of probiotics that may occur during spaceflight.

## Figures and Tables

**Figure 1 life-13-00727-f001:**
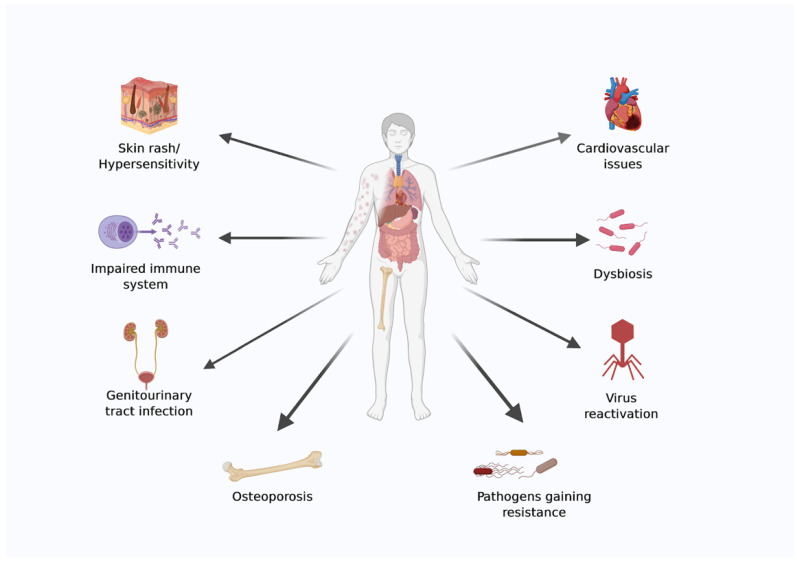
Diagrammatic representation of health issues faced by astronauts during spaceflight. The image was created using BioRender.com.

**Table 1 life-13-00727-t001:** The effect of microgravity on the gut microbiota of astronauts during spaceflight.

S. No	Study	Changes Observed during Spaceflight	Reference
1	The NASA Twin Study: one comparative study on the impact of spaceflight on one twin the twin served as genetically match ground control	The composition of the gut microbiota was significantly altered. There were significant differences in gene regulation and telomere length between the twin on Earth and the astronaut twin. Reduced anti-inflammatory effect of metabolites.	(Garrett-Bakelman et al., 2019), [21]
2	The impact of long-term spaceflight on the microbiome of nine astronauts who spent six to twelve months on the International Space Station (ISS) was studied.	During the space mission, microbial communities in the gastrointestinal tract, nose, skin, and tongue changed. The composition of the gut microbiota became more similar in astronauts in space. Latent virus reactivation	(Voorhies et al., 2019), [1]
3	Effect of short-term spaceflight on human microbiota (fecal sample study)	Short-term spaceflight did affect the composition and function of gut microbiota. Decrease in *Lactobacillus* and *Bifidobacterium* bacteria, while the increase in Bacteroides.	(Liu et al., 2020), [15]
4	STARMAPs Test (similarity in two space research database findings)	Vigorous change in mammalian gut microbiota due to spaceflight.	(Jiang et al., 2019), [23]
5	Lunar Palace 1 (Bliss Study), study of gut microbiota of astronauts and humans on Earth.	The gut microbiota of comparatively different between astronauts (significantly decreased) and humans because of different dietary and different lifestyles. Bliss also suggested the high-fibre diet could be beneficial for gut microbiota.	(Hao et al., 2018), [32]

**Table 2 life-13-00727-t002:** The immunological changes observed in astronauts during space flight.

Sr. no.	Duration of Space-Flight	No. of Individuals	Changes in Immune Cells	Reference
1.	5–11 days	25	An 85% increase in the granulocytes was observed as compared to pre-flight values with a significant reduction in phagocytosis and oxidative burst capacities.	Kaur et al., 2004, [48]
2.	5–11 days	25	A reduction in ability to engulf *E. coli*, oxidative burst and degranulation was elucidated by monocytes following spaceflight. A reduction in phagocytosis was observed with changes observed in the expression of surface markers.	Kaur et al., 2005, [53]
3.	10 days	4	The continuous production of immunoglobulins was prolonged in weightlessness and the process of lymphocyte activation may be impaired and hence altering the responses to new antigenic stimuli in microgravity conditions.	Voss, 1984, [54]
4.	4–16 days	11	A significant increase in the number of circulating WBCs, neutrophils, monocytes, T-helper cells and B cells. The number of NK cells decreased.	Mills et al., 2001, [55]
5.	8–15 days	16	A slight decrease in lymphocytes and a 1.5-fold increase in the neutrophil number with increased adhesion to the endothelial cells after spaceflight.	Stowe et al., 1999, [56]
6.	9–16 days	28	An increase in CD4+ cells, polymorphonuclear leukocytes and monocytes whereas there was a decrease in NK cells and monocytes after 9-day and 16-day spaceflight, respectively.	Stowe et al., 2003, [49]
7.	6 months	23	A reduction in the function of T-cells was observed with alterations in CD8+ cells. An increase in the number and redistribution of WBCs was observed.	Crucian et al., 2015, [14]

## Data Availability

Not applicable.

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
