# Peer review of "Prospective Use of Probiotics to Maintain Astronaut Health during Spaceflight"

_life, 2023, doi:10.3390/life13030727_

Round 1

Reviewer 1 Report

The manuscript overlooks the significance of mental health of the astronauts during the prolong space travel, and the important role of the gut-brain axis in regulating astronauts' cognition and behavior. I suggest that this important topic is at least mentioned.

The manuscript needs editing as there are several sections: 2.5.,2.6.,3.2.1, 3.3., that contain confusing sentences.

Sections 3.2.1 and 3.3. are very confusing, and the inividual points in these sections are not necessary.

There is no reference in the text to the Table 1 or Figure 1.

I suggest that the data on the microbiota and probiotic use in the astronauts is summarized in additional tables.

Reviewer 2 Report

The review from Bharindwal et al elaborates on a relevant topic of application of probiotic bacteria in maintenance of astronaut health. I only have minor points to suggest to improve the overall quality of this review. 

1. In section 1, Introduction, the authors talk about few gut microbiota members and their effects on host physiology. However as many other reviews summarize a plethora of gut-microbiota/probiotic bacteria beyond the ones described by the authors have effects on host physiology- hence this point will have to be mentioned. There are many other microbiota members with reported health benefits. Since it is difficult to summarize everything in this review, it would be nice to include a sentence referring to other reviews in the field and mentioning a general statement that a variety of different probiotic bacteria are known to confer health benefits to host physiology (and cite relevant references). Example of one review: https://pubmed.ncbi.nlm.nih.gov/19343057/. There are others too. 

2. In section 2.1, second paragraph, the appropriate word is "nasal" microbiota, not nose microbiota. 

3. In section 2.2, the sentence "Nonpathogenic and pathogenic strains of E. coli showed improved adherence and intrusion in vitro considering microgravity. " is not clear. Do you mean there was increased bacterial adherence under the influence of microgravity?

4. Across the manuscript, 4-5 times I found the bacterial names are not italicized. They must be italicized. 

5. In the section 2.5, second paragraph, what does "Proteins during infection with enterotoxigenic E.coli k88 (Yi et al., 2018)" mean? It's not understandable. 

6. Section 3.2 is an important section for this review. It should be as comprehensive as possible. For example, smg studies conducted on animal models ( in vivo) have to be included in addition to invitro studies. Upon quick research, I found a study fitting for this section (https://www.mdpi.com/2075-1729/12/11/1865#:~:text=Based%20on%20our%20findings%2C%20several,observed%20under%20simulated%20microgravity%20conditions.). Similarly, there could be other studies too. Please make the section as comprehensive as possible by including all the relevant information. 

Reviewer 3 Report

This review interestingly discusses some mechanisms associated with gastrointestinal dysbiosis in astronauts and the potential use of probiotics. The work is interesting and the review of the literature well-done.

I have just some minor suggestions that I hope can help the authors to better cover all the aspects associated to the presented work:

- The authors cite that "space fever" can be present, and with that, hyperthermia and fluid balance alterations. Even in absence of such a phenomenon, thermal stress and altered fluid balance are common in astronauts and both these factors are well-known to affect the microbiota. Can the authors better comment on this? For example, the use of probiotics has been discussed also to mitigate the effect of heat stress and exercise on gut microbiota, with conflicting results (Gill et al., 2016; Rauch et al., 2022).

- The gut-brain axis represents an interesting topic as it might help to identify possible crosstalk mechanisms between these two systems, and it might be interesting considering the high prevalence of neurophysiological and gut microbiota alterations (Buoite Stella et al., 2021); are you aware of any study linking these two systems in astronauts and space medicine?
